# Faithful Dynamic Imitation Learning from Human Intervention with Dynamic Regret Minimization

**Bo Ling**
Southeast University
`bo_ling@seu.edu.cn`

**Zhengyu Gan**
Southeast University
`zhengyugan@seu.edu.cn`

**Wanyuan Wang**
Southeast University
`wywang@seu.edu.cn`

**Guanyu Gao**
Nanjing University of Science and Technology
`gygao@njust.edu.cn`

**Weiwei Wu**
Southeast University
`weiweiwu@seu.edu.cn`

**Yan Lyu**[*]
Southeast University
`lvyanly@seu.edu.cn`

## Abstract

Human-in-the-loop (HIL) imitation learning enables agents to learn complex behaviors safely through real-time human intervention. However, existing methods struggle to efficiently leverage agent-generated data due to dynamically evolving trajectory distributions and imperfections caused by human intervention delays, often failing to faithfully imitate the human expert policy. In this work, we propose Faithful Dynamic Imitation Learning (FaithDaIL) to address these challenges. We formulate learning from human intervention as an online non-convex problem and employ dynamic regret minimization to adapt to the shifting data distribution and track high-quality policy trajectories. To ensure faithful imitation of human expert despite training on mixed agent and human data, we introduce an unbiased imitation objective and achieve it by weighting the behavior distribution relative to the human expert's as a proxy reward. Extensive experiments on MetaDrive and CARLA driving benchmarks demonstrate that FaithDaIL achieves state-of-the-art performance in safety and task success with significantly reduced human intervention data compared to prior HIL baselines. The corresponding source code is available at `https://github.com/William-island/FaithDaIL`.

## 1 Introduction

Human-in-the-loop (HIL) imitation learning is a promising approach to address key limitations of traditional reinforcement learning, such as misalignment with human intent and poor sample efficiency. By incorporating human feedback and interventions into the learning process, HIL enables agents to better align with human preferences, even when reward functions are difficult to design or prone to unintended biases [1, 2, 3]. While early HIL methods relied on passive human feedback – such as preference on behavior trajectories [4, 5, 6, 7] – these methods are unsuitable for safety-critical domains like autonomous driving, where unacceptable risks can arise during data collection [8, 9].

In contrast, active human intervention (see Figure 1a), where humans provide real-time corrections and demonstrations during agent execution, directly enhances training-time safety. While some early methods like Human-Gated DAgger (HG-DAgger) [8] focus solely on human-provided data, this data

---
[*]Corresponding Author

39th Conference on Neural Information Processing Systems (NeurIPS 2025).

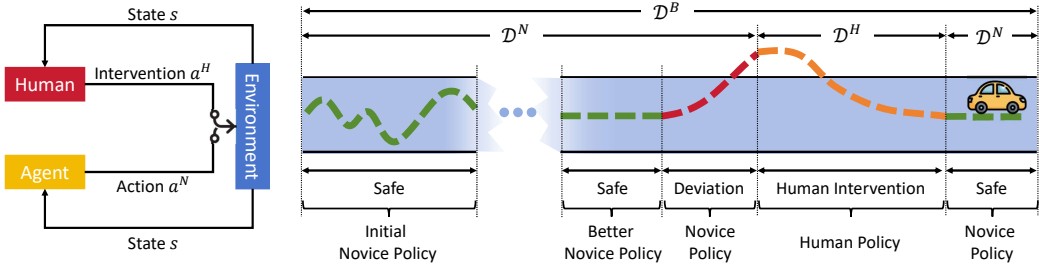

Figure 1: Learning from Intervention. a) Agent-Environment Interaction: human expert supervises the learning agent interacting with the environment, and intervenes when necessary. b) Agent Behavior Trajectories: includes safe but not optimal trajectory from initial suboptimal novice policy, improved novice policy trajectory, deviation (red) due to human reaction latency, human intervention and subsequent recovery. Our FaithDaIL leverages novice policy trajectories $\mathcal{D}^N$ together with human intervention $\mathcal{D}^H$ to improve data efficiency, while explicitly considering the evolving novice data distribution shifts and intervention latency to achieve faithful imitation of human experts.

is often costly and limited. To improve data efficiency, recent methods incorporate agent-generated trajectories (often called *novice trajectories*) as supplementary training data, combining them with human interventions in an off-policy imitation learning framework [9, 10, 11].

However, leveraging agent-generated trajectories in HIL introduces two challenges. First, as the agent's policy improves throughout training, the distribution of its generated trajectories shifts significantly (see green trajectories in Figure 1b). Using these trajectories indiscriminately – many originating from early, suboptimal policies – impedes effective policy updates if all historical data is treated equally. Our *first key insight is that learning from human intervention problem is fundamentally an online learning problem with a dynamically changing data distribution*. Instead of viewing it as a static problem with an expanding dataset like DAgger [12], we formulate it as a *dynamic regret minimization* problem to explicitly adapt to the evolving behavior distributions over time.

Second, even recent agent-generated data may be imperfect due to human reaction delays during interventions. The agent might execute incorrect actions just before an intervention (see red deviation trajectory in Figure 1b). Imitating these "deviation trajectories" risks teaching the agent suboptimal or unsafe behaviors. Our *second key insight is that while leveraging agent-generated data is crucial for data efficiency, the primary imitation target should be the human expert policy*. Therefore, we need an unbiased objective that ensures faithful imitation of the human expert, even when training on a mixture of expert and novice data.

In this paper, we propose Faithful Dynamic Imitation Learning, FaithDaIL, a novel HIL approach that enhances data efficiency by integrating evolving novice trajectories through dynamic regret minimization. It further achieves faithful imitation of human experts by deriving a proxy reward from weighting the mixed behavior distribution to the expert's distribution. To the best of our knowledge, this is the first work that formulate learning from human intervention as an online non-convex learning problem to minimize dynamic regret of imitation loss. In summary, our contributions are:

- We propose Faithful Dynamic Imitation Learning framework, FaithDaIL, that formally formulates learning from human intervention as an online non-convex learning problem and adopts dynamic regret as the performance metric to explicitly adapt to the changing data distribution induced by the evolving novice policy.

- We propose an unbiased objective for faithful human expert imitation from mixed data (novice trajectories and human interventions), and achieve it by weighting the behavior distribution relative to the human expert's as a proxy reward.

- We conducted extensive experiments on MetaDrive and CARLA driving benchmarks. Results show FaithDaIL significantly outperforms leading HIL baselines in safety and task success, with notably less human intervention data.

## 2 Related Work

**Human-in-the-loop Imitation Learning.** Many studies explore the incorporation of humans into the training loop in imitation learning. Passive approaches, such as DAgger [12] and its variants [13, 14, 15, 16, 17], address the compounding error problem [18] inherent in behavior cloning by periodically querying an expert for additional demonstrations on agent-visited states. Instead of providing demonstrations upon requests, active approaches allow experts to take control and guide the agent to safer states. Human-Gated DAgger (HG-DAgger) [8] involves human intervention during training and learns a policy by leveraging the collected human data. However, using only human data is costly and limited. To improve data efficiency, recent methods such as Expert Intervention Learning (EIL) [19], Intervention Weighted Regression (IWR) [10], HACO [11], and PVP [9] incorporate agent-generated trajectories as supplementary data, combining them with human interventions in an off-policy imitation learning framework. EIL [19] and IWR [10] treat agent-generated data in a supervised manner, while HACO [11] and PVP [9] leverage off-policy RL techniques to imitate a combination of agent-generated data and human data.

However, these methods face two major drawbacks: 1) they effectively minimize static regret on aggregate data, neglecting the evolving data distribution shifts, and 2) they imitate mixed behavior trajectories instead of focusing solely on the expert, leading to learning suboptimal actions caused by intervention latency (only partially addressed by EIL's fixed window, which cannot adapt to dynamic environment [19]). Our FaithDaIL addresses these by formulating human-in-the-loop learning as dynamic regret minimization to align with evolving distributions and proposing an unbiased off-policy imitation objective based on DICE [20] for faithful expert imitation.

**Off-policy Imitation Learning.** Human-in-the-loop (HIL) imitation learning, characterized by sparse and disruptive interventions, makes on-policy data collection inefficient; off-policy methods are therefore essential [21]. While behavior cloning (BC) [22] is inherently off-policy, it suffers from compounding errors and ignores environment dynamics [18]. For adversarial methods, DAC [21] extends GAIL [23] to off-policy setting using a replay buffer for better sample efficiency. The DICE family [20, 24, 25, 26] addresses off-policy imitation via stationary distribution matching. However, these methods primarily focus on expert demonstrations. Based on DICE, some methods incorporate offline supplementary imperfect demonstrations to enhance data efficiency [27, 28, 29, 30, 31] but rely on a strict coverage assumption [27, 29, 32]. In our FaithDaIL, by defining the behavior data as a combination of agent-generated and human expert data, the coverage requirement (i.e., the support of the expert distribution is covered by the behavior distribution) is naturally satisfied, enabling faithful imitation learning from human interventions.

**Online Non-convex Learning.** Online learning offers a principled framework for sequential decision making [33]. In the convex setting, Online Gradient Descent (OGD) and its variants achieve optimal sub-linear static regret, and many extensions bound dynamic regret [34, 35, 36, 37, 38, 39]. With non-convex losses—common in deep networks—regret minimization is NP-hard [40]. Follow-the-Perturbed-Leader (FTPL) algorithms [41, 42] achieve sublinear static regret for online non-convex learning (with an oracle), but are not well-suited for dynamic distributions. To address this, FTPL-D+ extends FTPL for dynamic environments using an ensemble and meta-algorithm [40]. In this work, we adopt FTPL-D+ to effectively manage the dynamic shifts in the novice agent's behavior policy.

## 3 Problem Formulation

We formulate learning from active human intervention with evolving policy trajectories as an online learning problem. Here, we first model agent-environment interaction as a Markov Decision Process, and then define the online learning problem as dynamic regret minimization.

The interaction between the environment and policy can be formulated as a Markov Decision Process (MDP), denoted by $\mathcal{M}$, and $\mathcal{M} = (\mathcal{S}, \mathcal{A}, p, r, \gamma, d_0)$, where $\mathcal{S}$ is the state space, $\mathcal{A}$ is the action space, $p$ is the transition probability function, $r : \mathcal{S} \times \mathcal{A} \to \mathbb{R}$ is the reward function, $\gamma \in (0, 1)$ is the discount factor, and $d_0$ is the distribution of initial state $s_0$. A policy $\pi(a|s)$ takes action $a$ in $\mathcal{A}$ given state $s$ in $\mathcal{S}$. Conventional Reinforcement Learning (RL) aims to find a policy $\pi$ that maximizes the expected return: $\mathbb{E}_\pi[\sum_{t=0}^{\infty} \gamma^t r(s_t, a_t)]$, where $\mathbb{E}_\pi$ denotes the expectation under the distribution induced by $a_t \sim \pi(\cdot|s_t)$, $s_{t+1} \sim p(\cdot|s_t, a_t)$. The distribution of a policy can be defined by its

discounted state-action visitation distribution $d_\pi(s,a) = (1-\gamma)\pi(a|s)\sum_{t=0}^{\infty}\gamma^t P(s_t = s|\pi)$. The unique stationary policy that induces a visitation $d(s,a)$ is given by $\pi(a|s) = d(s,a)/\sum_a d(s,a)$.

With human expert involvement, we denote the human expert policy as $\pi^H(a|s)$ and the learning agent's novice policy as $\pi^N(a|s)$. We also use a Boolean indicator $I(s, a^N) : \mathcal{S} \times \mathcal{A} \to \{0, 1\}$ to indicate when the expert intervenes ($I(s, a^N) = 1$). Thus, the actual action interacts with environment is $I(s, a^N)a^H + (1 - I(s, a^N))a^N$, with corresponding behavior policy $\pi^B$:

$$\pi^B(a|s) = I(s, a^N)\pi^H(a|s) + (1 - I(s, a^N))\pi^N(a|s).$$

Since the novice policy $\pi^N$ updates during training, its distribution $d^N$ changes over time. Consequently, the actual distribution $d^B$ of behavior policy $\pi^B$ also changes over time. This evolving dynamics motivate us to formulate the learning objective as *dynamic regret minimization* from an online learning perspective. Specifically, at each training round $i$, the novice policy $\pi_i^N$ interacts with the environment to collect data $\mathcal{D}_i^N$, while human expert intervenes with policy $\pi^H$, expanding the entire human expert dataset to $\mathcal{D}_i^H$. The combined dataset $\mathcal{D}_i^B = \mathcal{D}_i^N \cup \mathcal{D}_i^H$ can be viewed as being sampled from the distribution $d_i^B$ induced by the behavior policy $\pi_i^B$. The learning objective is to minimize the dynamic regret:

$$R_D = \sum_{i=1}^{M} \ell(\pi_i^N, \mathcal{D}_i^B, \mathcal{D}_i^H) - \sum_{i=1}^{M} \ell(\pi_i^*, \mathcal{D}_i^B, \mathcal{D}_i^H), \tag{1}$$

where $\ell$ denotes the imitation loss under the empirical mixture distribution estimated by sampling from $\mathcal{D}_i^B$ at round $i$ (see details in Sec 4.1), and $\pi_i^* = \arg\min_\pi \ell_i(\pi, \mathcal{D}_i^B, \mathcal{D}_i^H)$ is the best policy in hindsight at round $i$. Unlike static regret that assumes a static policy learned from all past data, this dynamic regret objective allows us to continuously compare the current policy with the best policy under the latest data, and thus capturing the dynamic of evolving trajectory distribution.

# 4 Method

FaithDaIL tackles learning from human intervention through two core components: an unbiased objective to ensure faithful imitation of the human expert, and an online non-convex learning algorithm that minimizes dynamic regret of the faithful imitation loss to adapt to evolving data distributions.

## 4.1 Faithful Imitation Objective with Behavior Trajectory

Previous methods that learn from human intervention typically perform off-policy imitation learning on the behavior distribution $d^B$, i.e., they imitate the actual trajectories from both expert and novice policy. However, some novice trajectories may be poor demonstrations, as human reaction delays during interventions can lead the agent to execute incorrect actions just before the human takes over. We therefore focus on faithfully imitating only the human expert, while still leveraging novice data for data efficiency.

We first define our imitation objective as minimizing Kullback-Leibler divergence from agent policy to human policy distribution $d^H(s,a)$, i.e.,

$$D_{\mathrm{KL}}\left(d^\pi(s,a) \,\|\, d^H(s,a)\right) = \mathbb{E}_{(s,a)\sim d^\pi}\left[\log \frac{d^\pi(s,a)}{d^H(s,a)}\right]. \tag{2}$$

Given that expert data is often sparse, prior works have incorporated suboptimal data to improve data efficiency [27, 28, 29, 30, 31]. However, these approaches typically rely on a coverage assumption that the suboptimal data visitation covers that of the expert [27, 29, 32], which is not always guaranteed [32]. To deal with this, we introduce behavior distribution $d^B$ and reformulate the objective as

$$D_{KL}(d^\pi(s,a)\|d^H(s,a)) = \mathbb{E}_{(s,a)\sim d^\pi}\left[\log \frac{d^\pi(s,a)}{d^B(s,a)} + \log \frac{d^B(s,a)}{d^H(s,a)}\right] \tag{3}$$

$$= \mathbb{E}_{(s,a)\sim d^\pi}\left[\log \frac{d^B(s,a)}{d^H(s,a)}\right] + D_{KL}(d^\pi(s,a)\|d^B(s,a)). \tag{4}$$

Since $d^B$ is a mixture of novice policy distribution $d^N$ and expert policy distribution $d^H$, we have $d^B > 0$ wherever $d^H > 0$. Therefore, the coverage requirement is satisfied at all times. To optimize this objective, we impose the Bellman-flow constraint $\sum_{a \in \mathcal{A}} d(s, a) = (1 - \gamma)d_0(s) + \gamma \sum_{s', a'} d(s', a')p(s|s', a')$ on states [43] and apply Lagrangian duality and convex conjugate [25]. This reformulates the intractable imitation objective into a tractable optimization over a value function, using a proxy reward derived from the ratio $d^B(s, a)/d^H(s, a)$. Specifically, we minimize:

$$V^\star = \arg \min_V (1 - \gamma)\mathbb{E}_{s \sim d_0} V(s) + \mathbb{E}_{(s,a) \sim d^B}[f^*(\mathcal{T}_{\tilde{r}}V(s, a) - V(s))], \tag{5}$$

where $V^\star$ denotes the optimal value, $f^*$ denotes the convex conjugate of the KL divergence, and $\mathcal{T}_{\tilde{r}}V(s, a) = \tilde{r}(s, a) + \gamma\mathbb{E}_{s' \sim p(s'|s,a)}[V(s')]$ is the Bellman operator defined using a proxy reward $\tilde{r}(s, a)$ given by:

$$\tilde{r}(s, a) = -\log\left[\frac{d^B(s, a)}{d^H(s, a)}\right] = -\log\left[\frac{1 - c^\star(s, a)}{c^\star(s, a)}\right],$$

where $c^\star$ is the optimal discriminator derived by:

$$\max_c \; \mathbb{E}_{(s,a) \sim d^H}[\log c(s, a)] + \mathbb{E}_{(s,a) \sim d^B}[\log(1 - c(s, a))].$$

Although the optimization for Eq. (5) doesn't contain the policy, it actually learns an implicit optimal policy through the visitation distribution ratio between the optimized and behavior policy. Therefore, we can extract the policy using weighted behavior cloning [44], where the optimal weight $\omega^\star(s, a)$ can be calculated by the ratio between the optimal policy distribution $d^\star(s, a)$ and $d^B(s, a)$, i.e.,

$$\omega^\star(s, a) = \frac{d^\star(s, a)}{d^B(s, a)} = \max\left(0, (f')^{-1}(\mathcal{T}_{\tilde{r}}V^\star(s, a) - V^\star(s))\right). \tag{6}$$

Then the optimal policy $\pi^\star$ can be found with the optimal weight $\omega^\star(s, a)$, i.e.,

$$\pi^\star = \arg \max_\pi \mathbb{E}_{(s,a) \sim d^B}[\omega^\star(s, a) \log \pi(a|s)]. \tag{7}$$

In summary, by first solving for the value function $V^\star$ via Eq. (5), and performing weighted behavior cloning using the derived weights $\omega^\star(s, a)$, we can learn the optimal policy $\pi^\star$.

**Empirical Faithful Imitation Objective.** In practice, we do not have access to the full distributions $d^H$ and $d^B$, so we sample from the datasets $\mathcal{D}_i^H$ and $\mathcal{D}_i^B$ collected at each training round $i$. The policy can be estimated by minimizing the empirical weighted behavior cloning loss, i.e.,

$$\ell(\pi_i^N, \mathcal{D}_i^B, \mathcal{D}_i^H) = -\mathbb{E}_{(s,a) \sim \mathcal{D}_i^B}\left[\omega_i^\star(s, a) \log \pi_i^N(a|s)\right], \tag{8}$$

where weight $\omega_i^\star(s, a)$ is derived from the optimized value function $V_i^\star$ by Eq. (6) in round $i$. This loss directly measures how well the novice policy $\pi_i^N$ imitates the human expert with the importance weighting. Theoretical derivations and implementation details can be found in Appendix A.

### 4.2 Learning from Evolving Behavior Trajectory with Dynamic Regret Minimization

Existing human-in-the-loop imitation learning methods either assume convexity and target static regret using algorithms like Follow-the-Leader (FTL) (e.g., DAgger [12], EIL [19]), or employ deep neural networks leading to non-convex optimization landscapes (e.g., HACO [11], PVP [9], IWR [10]). While the latter operate online, they often update policies by learning from the aggregate of all historical data, which is quite similar to the idea of optimizing for static regret using the Follow-the-Perturbed-Leader (FTPL) algorithm [42]. That is, they optimize policy on the accumulated historical data at every round $i$. However, static regret is insufficient as it overlooks the evolving quality of $\mathcal{D}^B$ and can bias the policy towards outdated trajectories.

To address this, we employ FTPL-D+ [40], an ensemble learning framework designed for non-convex online learning to optimize for dynamic regret (Eq. (1)). FTPL-D+ extends the classical Follow-the-Perturbed-Leader (FTPL) algorithm to non-stationary environments where the loss landscape evolves over time. It maintains a set of $K$ FTPL learners, each associated with a different rolling window size, to capture different temporal dynamics. This mechanism implicitly smooths out the non-stationarity of the data distribution and allows the overall strategy to adapt to varying timescales of change. Such

a multi-interval ensemble is particularly well-suited for minimizing dynamic regret, as it balances short-term adaptivity (via short-horizon learners) and long-term stability (via long-horizon learners). Theoretical results of FTPL-D+ [40] show that this structure achieves near-optimal dynamic regret bounds in general non-convex settings.

Building on this ensemble structure, each individual learner follows a standard restarting strategy: the time horizon $M$ is partitioned into intervals of length $\tau$, and FTPL is restarted at the beginning of each interval. Given a policy parameterization $\theta_\pi$, the update at round $i$ is:

$$\pi_{i+1}^N = \arg\min_\pi \left( \sum_{j=\mu_\tau}^i \ell(\pi, \mathcal{D}_j^B, \mathcal{D}_j^H) + \sigma_i^\top \theta_\pi \right), \tag{9}$$

where $\mu_\tau = \tau \lfloor (i-1)/\tau \rfloor + 1$ is the beginning of the interval, $\theta_\pi$ is the parameter of novice policy, and $\sigma_i \in \mathbb{R}^d$ is a random perturbation vector, whose components are typically sampled i.i.d. from an exponential distribution $\mathrm{Exp}(\eta)$ with parameter $\eta > 0$ at each round $i$.

A properly chosen $\tau$ allows the policy to track the optimal behavior distribution. Since the optimal $\tau$ is unknown, FTPL-D+ maintains an ensemble of $K$ base learners (novice policies $\{\pi_{i,k}^N\}_{k=1}^K$), each associated with a different interval parameter $\tau_k = 2^{k-1}$ for $k = 1, \ldots, K = \lfloor \log_2 M \rfloor + 1$. A meta-algorithm, Hedge [45], adaptively assigns weights $\alpha_{i,k}$ to each learner. At each round $i$, after observing new data $\mathcal{D}_i^B$ and $\mathcal{D}_i^H$, the weights are updated:

$$\alpha_{i+1,k} = \frac{\alpha_{i,k} e^{-\rho \ell(\pi_{i,k}^N, \mathcal{D}_i^B, \mathcal{D}_i^H)}}{\sum_{k'=1}^K \alpha_{i,k'} e^{-\rho \ell(\pi_{i,k'}^N, \mathcal{D}_i^B, \mathcal{D}_i^H)}}, \tag{10}$$

where $\rho > 0$ is a learning rate controlling the sensitivity of the weight update. A novice policy is then sampled according to the distribution $\{\alpha_{i+1,k}\}_{k=1}^K$ to interact with the environment in the next round. This enables dynamic adaptation to shifting behavior distributions.

Our Faithful Dynamic Imitation Learning (FaithDaIL) algorithm is presented in Algorithm 1. The algorithm firstly initializes the ensemble of $K$ novice policies, their adaptive weights $\alpha_{1,k}$ and interval parameter $\tau_k = 2^{k-1}$ (Lines 1-2). In each round $i$, a novice policy $\pi_{i,k'}^N$ is sampled based on the current weights $\{\alpha_{i,k}\}$ to interact with the environment (Line 4). During the interaction, a human expert provides real-time interventions when necessary. We collect new novice trajectory data $\mathcal{D}_i^N$

---

**Algorithm 1** Faithful Dynamic Imitation Learning from Human Intervention (FaithDaIL)

---

1: Set $K = \lfloor \log_2 M \rfloor + 1$, $\mathcal{D}_0^H = \varnothing$
2: Initialize $K$ novice policies $\{\pi_{1,k}^N\}_{k=1}^K$, weights $\alpha_{1,k} \leftarrow \frac{1}{K}$, interval parameter $\tau_k = 2^{k-1}$
3: **for** $i = 1$ to $M$ **do**                                    ▷ Online learning rounds
4:     Sample policy index $k' \sim \mathrm{Categorical}(\{\alpha_{i,k}\})$
5:     Execute $\pi_{i,k'}^N$ to interact with the environment and collect novice policy data $\mathcal{D}_i^N$
6:     Human expert provides real-time interventions, collect intervention data $\mathcal{D}_i^{H,\mathrm{new}}$
7:     Update $\mathcal{D}_i^H = \mathcal{D}_{i-1}^H \cup \mathcal{D}_i^{H,\mathrm{new}}$
8:     Construct behavior data $\mathcal{D}_i^B = \mathcal{D}_i^N \cup \mathcal{D}_i^H$
9:     **for** $k = 1$ to $K$ **do**                    ▷ Evaluate loss of each policy on current data
10:         Compute imitation loss $\ell(\pi_{i,k}^N, \mathcal{D}_i^B, \mathcal{D}_i^H)$ by Eq. (5), (6), and (8)
11:     **end for**
12:     Update ensemble weights based on Eq. (10)
13:     **for** $k = 1$ to $K$ **do**                    ▷ Update novice policies over each interval
14:         Let $\mu_{\tau_k} = \tau_k \lfloor (i-1)/\tau_k \rfloor + 1$
15:         Sample perturbation $\sigma_{i,k} \sim \mathrm{Exp}(\eta)$
16:         Update $\pi_{i+1,k}^N = \arg\min_\pi \left( \sum_{j=\mu_{\tau_k}}^i \ell(\pi, \mathcal{D}_j^B, \mathcal{D}_j^H) + \sigma_{i,k}^\top \theta_\pi \right)$
17:     **end for**
18: **end for**
19: **return** Policy $\pi_{M+1,k^\star}^N$, where $k^\star = \arg\max_k \alpha_{M+1,k}$

---

Table 1: Comparison of different approaches in MetaDrive-Keyboard and CARLA-Wheel.

| Method | MetaDrive-Keyboard | | | | | | CARLA-Wheel | | | |
| | Training | | | Testing | | | Training | | Testing | |
| | Human Data | Total Data | Total Safety Cost | Episodic Return | Episodic Safety Cost | Success Rate | Human Data | Total Data | Route Comp. | Success Rate |
|---|---|---|---|---|---|---|---|---|---|---|
| PPO | - | 1M | 26.4K | 327.33 | 3.31 | 0.76 | - | 1M | 0.24 | 0.0 |
| TD3 | - | 1M | 1.90K | 317.45 | **1.44** | 0.58 | - | 1M | 0.11 | 0.0 |
| Human | - | - | - | 374.73 | 0.39 | 0.98 | - | - | 0.99 | 1.0 |
| BC | 30K | - | - | 129.60 | 17.40 | 0.12 | 5K | - | 0.42 | 0.20 |
| HG-DAgger | 7.5K | 30K | 143 | 297.60 | 7.05 | 0.59 | 6.8K | 24K | 0.64 | 0.47 |
| IWR | 6.1K | 30K | 112 | 327.32 | 9.16 | 0.75 | 5.7K | 24K | 0.69 | 0.60 |
| HACO | 9.9K | 30K | 76 | 239.41 | 4.29 | 0.26 | 4.8K | 24K | 0.52 | 0.40 |
| PVP | 7.0K | 30K | 54 | 343.86 | 2.51 | 0.85 | 6.6K | 24K | 0.92 | 0.73 |
| FaithDaIL | **4.8K** | 30K | 55 | **354.35**$_{\pm3.43}$ | 1.47$_{\pm0.28}$ | **0.91**$_{\pm0.04}$ | **4.2K** | 24K | **0.95**$_{\pm0.02}$ | **0.91**$_{\pm0.03}$ |

Table 2: Ablation Study in MetaDrive-Keyboard and CARLA-Wheel.

| Method | MetaDrive-Keyboard | | | CARLA-Wheel | |
| | Episodic Return | Episodic Safety Cost | Success Rate | Route Comp. | Success Rate |
|---|---|---|---|---|---|
| FaithDaIL w/o DRM | 346.06 $_{\pm5.42}$ | 2.29 $_{\pm0.29}$ | 0.87 $_{\pm0.04}$ | 0.91 $_{\pm0.03}$ | 0.81 $_{\pm0.01}$ |
| FaithDaIL w/o FOP | 350.26 $_{\pm3.57}$ | 1.78 $_{\pm0.51}$ | 0.89 $_{\pm0.05}$ | 0.86 $_{\pm0.04}$ | 0.73 $_{\pm0.07}$ |
| FaithDaIL (Ours) | **354.35** $_{\pm3.43}$ | **1.47** $_{\pm0.28}$ | **0.91**$_{\pm0.04}$ | **0.95** $_{\pm0.02}$ | **0.91** $_{\pm0.03}$ |

and accumulate intervention data $\mathcal{D}_i^{H,\text{new}}$ into $\mathcal{D}_i^H$ (Lines 5-7). The behavior data $\mathcal{D}_i^B$ is constructed by merging the novice data and human intervention data, i.e., $\mathcal{D}_i^B = \mathcal{D}_i^N \cup \mathcal{D}_i^H$ (Line 8). We then compute the imitation loss of each novice policy by Eq. (5), (6), and (8) (Lines 9-11). The ensemble weights for the next round is updated by Eq. (10) (Line 12). Each novice policy $\pi_k^N$ is updated via FTPL using data from its corresponding interval, defined by $\mu_{\tau_k} = \tau_k \lfloor (i-1)/\tau_k \rfloor + 1$. The update solves a regularized imitation loss minimization over the data from rounds $j = \mu_{\tau_k}$ to $i$, with exponential perturbation $\sigma_{i,k}$ (Lines 13–17). Finally, after $M$ rounds of training, FaithDaIL returns the novice policy with the highest ensemble weight (Line 19).

## 5 Experiments

### 5.1 Experimental Setting

**Environments.** We evaluate our approach on two challenging driving simulators: MetaDrive Safety Benchmark [46] and CARLA Town01 [47]. MetaDrive is a lightweight simulator in which agents must navigate vehicles safely in dense traffic. We created training and testing sets of 100 distinct scenarios to assess generalization. CARLA, a widely used autonomous driving platform, provides realistic urban settings in Town01 with low-level continuous control (acceleration, braking, steering). To evaluate robustness to different observation modalities, we use sensory state vectors in MetaDrive and bird's-eye view images in CARLA. Further details are in Appendix C.

**Evaluation metrics.** In MetaDrive, we use *episodic return* (cumulative reward per episode), *safety cost* (simulator-defined safety score), and *success rate* (reaching destination). Since CARLA does not provide reward or safety scores, we use *route completion* (distance traveled / total route length) and *success rate*. We also report *human data* and *total data* usage for training imitation algorithms.

**Experiment Procedure.** Three college students (aged 20–25 with valid driver's licenses) participated in the human-in-the-loop experiments. Prior to training, they practiced until proficient (achieving $\geq 95\%$ success over 50 episodes). Each participant then supervised a learning agent for approximately one hour per simulator, intervening via a keyboard in MetaDrive (Appendix C, Fig. C.1) or a Logitech

G923 racing wheel in CARLA (Appendix C, Fig. C.2) upon the agent risking safety/traffic violations or significantly deviating from human norms. Training scenarios were randomly selected from 100 diverse environments in MetaDrive and 25 varied routes with different start/end points, lighting, and weather conditions in CARLA Town01.

**Implementation Details.** Our implementation builds upon open-source repositories of ODICE [25], PVP [9] and FTPL-D+ [40]. Implementations of PPO and TD3 utilize Stable-Baselines3 [48], while other HIL baselines use official implementations where available [9, 11]. During testing, agents operate autonomously without human intervention. Experiments were run on a machine with an Nvidia GeForce RTX 3070 Ti Laptop GPU and an Intel Core i7-12700H CPU, supporting real-time simulation and training. Hyper-parameter and other details are in Appendix D.

## 5.2 Baseline Comparison

**Baselines.** We compare our method with standard RL baselines of PPO [49] and TD3 [50]. We also collected 30K high-quality human expert demonstrations with a $98\%$ success rate to train a Behavior Cloning (BC) policy [22]. For human-in-the-loop baselines, we compare with Human-Gated DAgger (HG-DAgger) [8], Intervention Weighted Regression (IWR) [10], Human-AI Copilot Optimization (HACO) [11], and Proxy Value Propagation (PVP) [9].

**Performance Comparison**. Table 1 summarizes the performance in MetaDrive and CARLA. RL methods (PPO, TD3), while capable of learning, require vast amounts of data (1M steps) and incur significantly higher total safety costs during training compared to HIL methods. This highlights the benefit of human guidance in reducing unsafe exploration and improving sample efficiency.

Among HIL methods, our FaithDaIL consistently outperforms baseline methods. In MetaDrive (Table 1, Left), it achieves the highest episodic return and success rate, and the lowest episodic safety cost, despite using the least human intervention data (4.8K steps, 16% of total). HG-DAgger performs poorly due to heavy reliance on limited human data. IWR improves with policy data but suffers from errors in early suboptimal trajectories. HACO shows lower testing safety cost but underperforms in task success, potentially due to inaccurate value estimation. PVP performs well among baselines but imitates the mixed behavior policy, making it susceptible to novice mistakes.

Similar trends are observed in the more visually complex CARLA environment (Table 1, Right), where FaithDaIL achieves the highest route completion and success rates with the fewest human interventions. These results underscore FaithDaIL's strength: by performing faithful imitation to expert and dynamically tracking high-quality data segments, it unbiasedly learns the human policy while avoiding error propagation from self-generated trajectories.

## 5.3 Ablation Study

We conducted ablation studies to assess effectiveness of FaithDaIL's key components: Dynamic Regret Minimization (DRM) and the Faithful Off-policy imitation learning module (FOP).

- FaithDaIL w/o DRM: Removes the FTPL-D+ ensemble framework, reducing the algorithm to training on all historical data with the FOP objective (akin to static regret minimization).
- FaithDaIL w/o FOP: Replaces the DICE-based faithful imitation objective (Eqs.(7)- (8)) with standard behavior cloning on the mixed novice/expert data, but with the DRM component.

Table 2 summarizes performance comparison. We observe that removing either DRM or FOP leads to performance degradation, especially in *episodic return* and *safety cost* in MetaDrive, and *route completion* and *success rate* in CARLA. Without DRM, training on all historical data (akin to static regret) impedes adaptation to the improving policy and complicates density ratio estimation for FOP. Without FOP, directly imitating mixed data exposes the agent to suboptimal novice actions due to intervention latency, preventing faithful expert imitation. These results confirm benefits of combining dynamic regret minimization and a faithful imitation objective.

## 5.4 Case Studies

To qualitatively evaluate performance, we conducted MetaDrive case study analyzing agent testing trajectories under six distinct road conditions. Figure 2 illustrates bird's-eye view snapshots of agents'

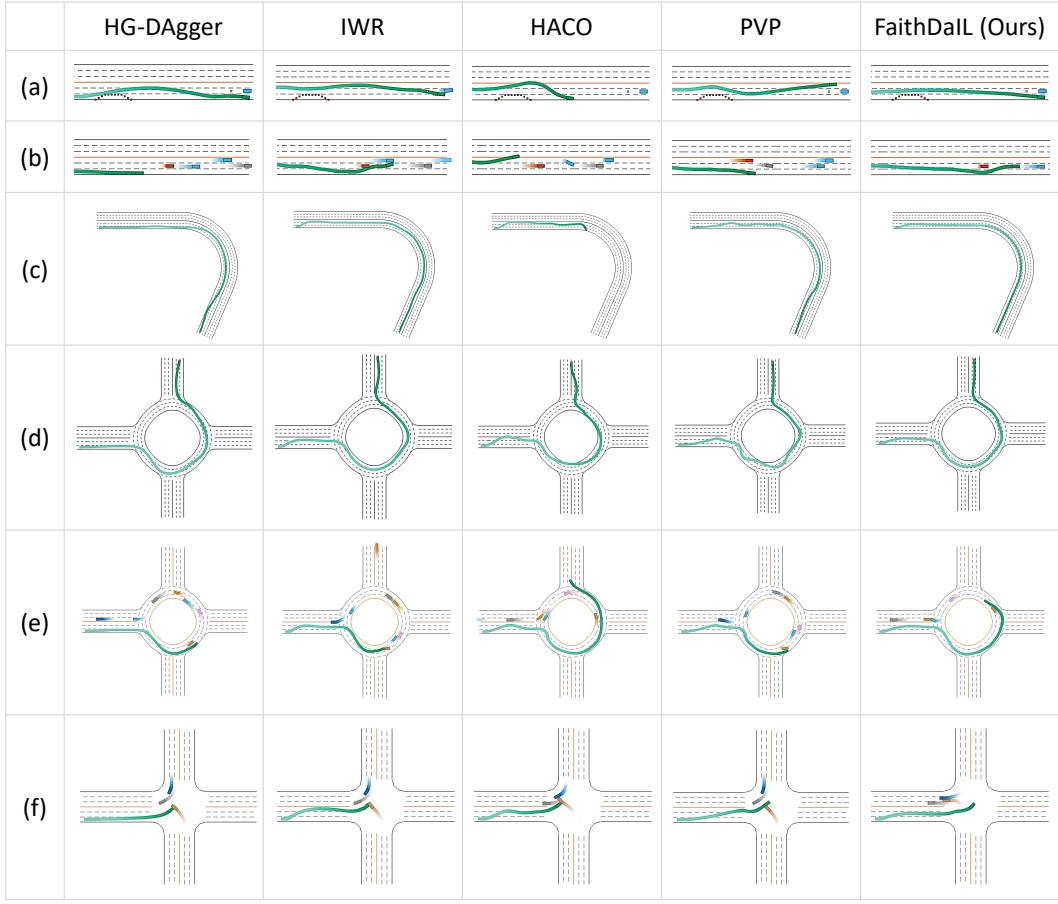

|  | HG-DAgger | IWR | HACO | PVP | FaithDaIL (Ours) |
|---|---|---|---|---|---|
| (a) | | | | | |
| (b) | | | | | |
| (c) | | | | | |
| (d) | | | | | |
| (e) | | | | | |
| (f) | | | | | |

Figure 2: Qualitative comparison of agent trajectories generated by different Human-in-the-Loop (HIL) methods in MetaDrive scenarios. Each row depicts a specific road condition: (a) straight road with static obstacles; (b) straight road with dynamic vehicles; (c) simple curve; (d) roundabout; (e) roundabout with dynamic vehicles; and (f) intersection with dynamic vehicles. The agent is indicated by the green rectangle, its recent path is traced by the green line, and other dynamic vehicles are represented by colored boxes. We can see that both HG-DAgger and IWR fail in scenarios with static and dynamic obstacles—HG-DAgger goes off-road in (b,e) and collides in (a,f), while IWR collides in (a,b,e,f)—and both exhibit erratic trajectories on curving roads (c,d). HACO performs even worse, showing unstable behaviors such as wide swings in (c,d,e) and lane departures in (a,b). PVP showed risky maneuvers to avoid obstacles, resulting in collisions in (f) and deviations in (a,b,e). In contrast, FaithDaIL produces the smoothest and safest trajectories, closely matching expert driving.

behaviors across these scenarios: (a) straight road with static obstacles; (b) straight road with dynamic vehicles; (c) simple curve; (d) roundabout; (e) roundabout with dynamic vehicles; and (f) intersection with dynamic vehicles. Videos are provided in the supplemental materials.

**Performance on Straight Roads with Obstacles (Static and Dynamic).** On straight roads with static obstacles (Figure 2a) or dynamic vehicles (Figure 2b), most baselines exhibited critical failures. HG-DAgger overreacted, causing collisions or off-road deviations. IWR collided with both static and dynamic elements. HACO showed unstable behavior like wide swings and lane departures. PVP showed risky maneuvers to avoid obstacles. In contrast, FaithDaIL consistently demonstrated smooth, safe navigation, avoiding all obstacles and adeptly handling dynamic traffic.

**Performance on Curves and Roundabouts (Without and With Dynamic Vehicles).** On curves and roundabouts without dynamic vehicles (Figure 2c, d), most baselines completed the trip but FaithDaIL provided the smoothest, most stable trajectories, closely resembling expert driving. With

dynamic vehicles (Figure 2e), all baselines failed (veering off-road or colliding). FaithDaIL was the only method to successfully and safely navigate around dynamic vehicles while staying on the lane.

**Performance at Intersections with Dynamic Vehicles.** Navigating intersections with oncoming traffic (Figure 2f) proved to be the most challenging scenario for the baseline methods. HG-DAgger, IWR, and HACO all resulted in collisions with incoming vehicles due to failure to react appropriately. PVP also failed to avoid a collision, being struck by another car. Our method was the only one to exhibit safe, intelligent behavior; it slowed down, yielded, and proceeded safely through the intersection, avoiding any collision or potential danger.

Overall, these case studies highlight the superior robustness and safety of our proposed method across a variety of challenging road conditions, particularly in complex dynamic environments where baseline HIL approaches frequently failed.

## 6 Conclusion and Future Work

In this paper, we proposed FaithDaIL, a Faithful Dynamic Imitation Learning framework to learn from human intervention. FaithDaIL incorporates policy-generated trajectories to improve data efficiency, while addressing challenges of dynamic data distribution shifts and trajectory deviations caused by intervention latency. FaithDaIL is the first work that formulates the learning as online non-convex optimization with dynamic regret minimization, thereby is able to track high-quality policy trajectories. We also proposed an unbiased objective for faithful expert imitation from mixed agent/human data via a weighted DICE-based approach. Extensive experiments on MetaDrive and CARLA demonstrated FaithDaIL achieves state-of-the-art safety and task success with significantly reduced human intervention.

Future work will address remaining challenges, including evaluation on realistic platforms (e.g., robotic arms) with more participants, improving robustness to diverse and suboptimal human interventions, and theoretical analysis on imitation capability of our method with dynamic regret minimization. We also aim to generalize FaithDaIL to broader HIL learning problems (e.g., multi-modal intervention).

## Acknowledgment

This work was supported in part by the Natural Science Foundation of China under Grant 62232004, 62572120, 62472093, 62572246; in part by the Key Research and Development Projects of Jiangsu Province (No.BE2021001-2); in part by the Natural Science Foundation of Jiangsu Province under Grant (No.BK20230024); in part by the Fundamental Research Funds for the Central Universities; and in part by the Big Data Computing Center of Southeast University.

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
