# OpenReview forum: "Faithful Dynamic Imitation Learning from Human Intervention with Dynamic Regret Minimization"
_NeurIPS.cc/2025/Conference — NeurIPS 2025 poster_

### Official Review · Reviewer_DSG8 · 2025-07-02

**Clarity:** 2
**Significance:** 2
**Originality:** 2
**Rating:** 4
**Confidence:** 4

**Summary:**

This paper proposes FaithDaIL, a new Human-in-the-Loop (HIL) imitation learning framework that integrates dynamic regret minimization (via FTPL-D+) and an unbiased imitation objective inspired by DICE to faithfully learn from human interventions. The method is evaluated on MetaDrive and CARLA, and empirically outperforms existing HIL baselines in safety and task success.

**Questions:**

[Q1] What is the fundamental novelty in FaithDaIL beyond combining DICE and FTPL-D+?

[Q2] What is the underlying rationale for FTPL-D+ being capable of solving the online IL problem well?

[Q3] Can the method generalize to non-driving domains?

[Q4] How sensitive is the method to the choice of time windows in FTPL-D+?

Minor comment: It is better to formally define the meaning of "faithful imitation" in the paper.

**Ethical Concerns:**

["NO or VERY MINOR ethics concerns only"]

**Final Justification:**

After rebuttal, the authors have addressed my concerns, and I increase my score to 4.

**Limitations:**

The paper lacks a thorough discussion of potential limitations and corresponding solutions.

**Quality:**

2

**Strengths And Weaknesses:**

**Strengths:**

[S1] The work addresses a meaningful problem: improving data efficiency and safety in HIL imitation learning.

[S2] The paper contributes some new insights to the community.

[S3] The empirical results on driving benchmarks are competitive.

**Weaknesses:**

[W1] The key introduced technique - DICE - is well-established in the literature, like [1]. Also, the combination of DICE and FTPL-D+ does not introduce a fundamentally new learning principle; the methodological contribution is more of a composition of known tools than an innovation.

[W2] There is a missing explanation for why FTPL-D+ can solve the online learning problem well. The paper does not provide new theoretical results, convergence analysis, or regret bounds tailored to the proposed problem. The use of dynamic regret is borrowed directly from prior work, and its connection to IL is not formalized beyond applying existing techniques.

[W3] The evaluation is mainly limited to driving simulators. There is no discussion or demonstration of applicability to other HIL settings (e.g., robotics, dialogue systems), making it unclear whether the method generalizes.

**Reference:**

[1] How to Leverage Diverse Demonstrations in Offline Imitation Learning. ICML. 2024.

---

> ### Author Rebuttal · Authors · 2025-07-31
>
> ### W1 & Q1: Novelty Clarification
>  We clarify our key contribution to the community of human-in-the-loop imitation learning (HIL-IL) are:
>
> This is the *first work to formulate HIL-IL as an online non-convex learning problem with dynamic regret as the optimization objective*.  Existing HIL-IL methods either use static regret or treat all imitation data  indiscriminately. We are *the first to identify the problem of shifting data distributions from evolving novice policy*, which has been totally overlooked in HIL-IL.
>
>  We are also *the first to introduce the faithful imitation objective for leveraging hybrid data from both experts and the learning policy*.  Existing methods (e.g., HACO, IWR, PVP) optimize on mixed novice-expert data without distinction, hindering performance by causing the agent to imitate suboptimal novice actions.
>
> We solve the dynamic regret problem using FTPL-D+ to adapt the dynamic non-convex loss and encode the faithful imitation objective into an unbiased proxy reward using DICE. These two techniques can be replaced with other methods (if any) as long as  they can slove dynamic regret problem with non-convex loss  and can optimize the unbiased imitation objective.
>
> In addition, it is true that using a discriminator to estimate a density ratio and using weighted behavior cloning to extract policy is a common technique in IL. However, the key difference is how the weights are computed. ILID [1] (mentioned by the reviewer) trains a discriminator to identify expert states and selects preceding actions based on trajectory backtracking. It requires access to contiguous trajectories, while in our human intervention data consists of fragmented trajectories from experts.
>
>
>
>
>
>
>
>
>
>   ### W2 & Q2: Rationale of using FTPL-D+ & Theoretical Analysis
>  The FTPL-D+ algorithm is an extension of the FTPL method. While FTPL is effective for optimizing static regret, it is not suited for dynamic distributions where environmental variation is unknown, a typical scenario in online imitation learning problems.
>
>  FTPL-D+ addresses this challenge by using an ensemble learning framework. It maintains a collection of learners, each associated with a different interval parameter, and partitions the time horizon to restart FTPL periodically. A meta-algorithm, Hedge, is then used to adaptively select among these learners. This enables the method to adapt to environmental changes and achieve a strong regret bound $\mathcal{O}\left(T^{2/3}(V_T + 1)^{1/3}\right)$, thereby being able to solve the online IL problems.
>
>
>  We will provide the following theoretical analysis on regret guarantees. Our theoretical analysis shows that our algorithm is no-regret and can eventually converge to the optimal policy：
>
> (Rendering issues with long equations led us to split them and add new notations (e.g., $k = i-1$), slightly reducing readability.)
>
> We aim to prove that our FaithDaIL is no-regret, i.e.,
>
> $
> \frac{1}{M} \mathbb{E}[\mathcal{R}_D] \to 0
> $
>
>
> We first give some assumption:
>
> + **Bounded Parameter Space**: The policy parameter space is effectively bounded due to regularization and optimizer constraints in practice.
> + **Lipschitz Loss**: The imitation loss $\ell_i(\pi)$ is Lipschitz in $\theta_\pi$, assuming (1) $\pi(a|s)$ is Lipschitz, (2) importance weights $\omega_i^\star(s,a)$ are bounded, and (3) $\pi(a|s)$ is bounded away from zero.
>
> Following the analysis from FTPL-D+, the **expected dynamic regret** of FaithDaIL satisfies:
>
> $
> \mathbb{E}[\mathcal{R}_D] \le \mathcal{O}\left( M^{\frac{2}{3}} (V_M + 1)^{\frac{1}{3}} \right),
> $
>
> where:
>
> - $M$ is the number of rounds,
> - $V_M =  \sum_{i=2}^{M} \max_{\pi \in \mathcal{K}} |l(\pi, \mathcal{D}_i^B, \mathcal{D}_i^H) - l(\pi,\mathcal{D}_k^B, \mathcal{D}_k^H)|$, $k=i-1$ is the total variation of the loss sequence.
>
> To establish that the dynamic regret is sublinear, **it suffices to show that $V_M$ grows sublinearly with $M$**. We formalize the implicit intervention rate $\beta_i$ and state the core assumption underlying its use:
>
> + **Definition (Implicit Intervention Rate $\beta_i$)** :
>
> $
> \beta_i = \mathbb{E}_{\pi^N_i } \left[ \frac{|\mathcal{D}_i^{H,\text{new}}|}{|\mathcal{D}_i^{H,\text{new}}| + |\mathcal{D}_i^N|} \right]
> $
>
> + **Assumption (Intervention Rate Decay)** : This reflects the intuition that as the agent improves, fewer human interventions are needed over time.
>
> $
> \lim_{i \to \infty} \beta_i = 0
> $
>
> To relate $\beta_i$ to the trajectory distribution shift, we invoke the following result (adapted from DAgger, Lemma 4.1):
>
> + **Lemma (Distribution Divergence Bound)**: With $T$ as the MDP horizon,
>
> $
> \|d_i^B - d_i^N\|_1 \le 2T\beta_i
> $
>
> We now bound the single-step loss variation $\Delta_i = \max_{\pi} | \ell_i(\pi) - \ell_{i-1}(\pi) |$.
>
> $
> |\ell_i(\pi) - \ell_{i-1}(\pi)|
> \leq |A - B| + |B - C|
> $
>
> where $k=i-1$ :
>
> $
> A = \mathbb{E}_{(s,a) \sim d_i^B}[w_i^\star(s,a) \log \pi(a|s)]
> $
>
> $
> B = \mathbb{E}_{(s,a) \sim d_k^B}[w_i^\star(s,a) \log \pi(a|s)]
> $
>
> $
> C = \mathbb{E}_{(s,a) \sim d_k^B}[w_k^\star(s,a) \log \pi(a|s)]
> $
>
>
>
> 1. **Term 1 (due to distribution shift):**：
>
> $
>    \left| A-B \right| = \left| \int_{(s,a)\in\mathcal{S}\times\mathcal{A}}[d_{i}^B(s,a) - d_{k}^B(s,a)] w_i^\star(s,a) \log \pi(a|s) ds  da  \right|
> $
>
> $
> \leq N_1 \cdot N_2 \cdot \|d_i^B - d_k^B\|_1
> $
>
> where:
>
> $
> N1 = \|w_i^\star \|_{\infty}
> $
>
> $
> N2 = \|\log \pi\|_{\infty}
> $
>
> 2. **Term 2 (due to weight shift):**：
>
> $
>    \left| B-C \right| = \left| \int_{(s,a)\in\mathcal{S}\times\mathcal{A}}[w_{i}^\star(s,a) - w_{k}^\star(s,a)] d_k^B(s,a) \log \pi(a|s) ds  da  \right|
> $
>
> $
> \leq N_2 \cdot N_3 \cdot \|w_{i}^\star - w_{k}^\star\|_1
> $
>
> where:
>
> $
> N2 = \|\log \pi\|_{\infty}
> $
>
> $
> N3 = \|d_k^B \|_{\infty}
> $
>
> **Step 1: Bounding distribution shift:** Using triangle inequality and Lemma (Distribution Divergence Bound):
>
> $
> \|d_i^B - d_k^B\|_1 \le 2T(\beta_i + \beta_k) + \|d_i^N - d_k^N\|_1
> $
>
> Assuming the novice policy changes smoothly:
>
> $\sum_{i=2}^M \|d_i^N - d_{i-1}^N\|_1 = o(M)$
>
>  and $\beta_i \to 0$, we have:
>
> $
> \sum_{i=2}^M \|d_i^B - d_{i-1}^B\|_1 = o(M)
> $
>
> **Step 2: Bounding weight shift:** Assuming that the variation in importance weights is primarily induced by the shift in behavior distributions, and that $V_i^\star$ evolves smoothly under $d_i^B$, we have:
>
> $
> \|w_i^\star - w_{i-1}^\star\|_1
> $
>
> $
> \le L_w \cdot \|d_i^B - d_{i-1}^B\|_1 + o(1)
> $
>
> Thus:
>
> $
> \sum_{i=2}^M \|w_i^\star - w_{i-1}^\star\|_1 = o(M)
> $
>
> Then under the earlier Lipschitz assumptions, we assume relevant terms of N1, N2, N3 are uniformly bounded.
>
> **Conclusion: Sublinear functional variation.**
>
> $
> V_M = \sum_{i=2}^M | \ell_i(\pi) - \ell_{i-1}(\pi) | = o(M)
> $
>
> **Final Result: No-regret Guarantee**
>
> From FTPL-D+ analysis:
>
> $
> \mathbb{E}[\mathcal{R}_D] = \mathcal{O}(M^{2/3}(V_M + 1)^{1/3}) = o(M), \quad
> \frac{1}{M} \mathbb{E}[\mathcal{R}_D] \to 0
> $
>
> Hence, **FaithDaIL is a no-regret algorithm** in non-convex dynamic environments, with the learned policy converging to the round-wise optimal.
>
>
>
>
>
>   ### W3 & Q3: Applicability to Other HIL Settings
> We will add a section of discussion about applicability to other HIL settings as follows.
>
> The core challenges FaithDaIL addresses—data efficiency and the impact of suboptimal novice data—are prevalent in many safety-critical domains beyond autonomous driving.
>
> Robotic Manipulation: In tasks like robotic arm manipulation, a human teleoperator may intervene to prevent errors. These interventions, coupled with the robot's evolving policy and potential reaction delays, create the exact type of dynamic, mixed-quality dataset that FaithDaIL is designed to handle. We plan to extend our work to this area using simulators to tackle tasks such as pick-and-place and object manipulation.
>
> Dialogue Systems: High-stakes dialogue systems for customer service or medical consultation also rely on human oversight. An operator may correct or override the system's responses, leading to a mix of optimal and suboptimal conversational paths. As the dialogue agent learns and its response distribution shifts, FaithDaIL's dynamic regret framework can be applied to improve its performance and reduce the need for intervention over time.
>
>
>
>
>
>
>   ### Q4: Time Window Sensitivity
>
>  FTPL-D+ employs an ensemble of base learners with different time windows ($\tau_k = 2^{k-1}$ where $k$ denotes the $k$-th learner) and leverages a Hedge-style meta-algorithm to adaptively adjust their weights based on performance. This design inherently mitigates the sensitivity problem to different window settings. In practice, we observe that during training, the meta-algorithm tends to assign more weight to policies with shorter time windows in later rounds, suggesting that the method self-adjusts to favor recent, high-quality behavior distributions.
>
>
>
> ### Q5:  Faithful Imitation Definition
>
>  "faithful imitation" refers to explicitly imitating the human expert's policy, even when using a mixed dataset of expert and novice trajectories for efficiency. This is accomplished through our imitation objective (Equations 2-3), which optimizes the KL divergence towards the human policy distribution. We will formally define this concept in the next version of the paper.
>
>
>
>
>
>
> ### Q6: Discussion on Limitation
>   We will discuss limitations as follows:
>   - Evaluation Scope: Validate the framework on other safety-critical platforms, such as robotic arms, by collecting and evaluating data from human teleoperator interventions.
>   - Expert Variability: Improve the model's robustness to diverse and potentially suboptimal interventions from different human experts. We can learn weight for expert data based on individual performance.
>   - Intervention Modality: Generalize the framework to support multi-modal feedback (e.g., verbal commands, gestures) instead of only direct control. This would involve encoding varied inputs into a consistent format for the learning loop.

---

> > ### Comment · Reviewer_DSG8 · 2025-08-07
> >
> > Thanks for the detailed response. After reading all the review comments and the authors' supplemental justification, most of my concerns have been addressed, but I'm still not fully convinced about the technical novelty of the proposed solution to the dynamic regret problem. Could the authors provide further clarification?

---

> > > ### Author Response · Authors · 2025-08-07
> > >
> > > We thank the reviewer for the positive feedback on our responses. For the technical novelty, we clarify that:
> > >
> > > 1) The key novelty of our work is not proposing a new algorithm for minimizing dynamic regret, but identifying and formally framing Human-in-the-Loop Imitation Learning (HIL-IL) as an online non-convex learning problem with non-stationary data. We are the first to formulate this as a dynamic regret minimization task. This approach distinguishes our work from prior methods that either treated HIL-IL as an online convex learning problem (e.g., DAgger, EIL) or overlooked the evolving data distributions (e.g., HACO, PVP).
> > >
> > > 2) We utilized and adapted the existing FTPL-D+ to solve this dynamic regret problem. A critical challenge arises here: the original FTPL-D+ is designed for online non-convex learning, where data streams generally come externally. However, in our HIL-IL setting, the agent must actively collect data by interacting with the environment. This distinction motivated our technical innovation: repurposing the Hedge-based sampling mechanism within FTPL-D+ as a novel, adaptive policy-switching mechanism for data collection. This process works as follows: a) FTPL-D+ maintains an ensemble of base learners. We use Hedge to select the best-performing learner to act as the "novice" policy for environmental interaction. b) This policy collects new data, which is then combined with expert data to retrain the entire ensemble. c) This creates a policy-data interaction loop that naturally stabilizes the learning landscape. Since the policy trained in one round generates the data for the next, the data distribution evolves gradually and self-consistently. This is a significant advantage over conventional online learning, where stability can be compromised by uncontrollable external data streams.
> > >
> > > 3) We also provided a new theoretical analysis of the dynamic regret bound within our HIL-IL framework. The original FTPL-D+ provides a general sublinear regret bound:
> > >
> > > $
> > > \mathbb{E}[\mathcal{R}_D] \le \mathcal{O}\left( M^{\frac{2}{3}} (V_M + 1)^{\frac{1}{3}} \right)
> > > $
> > >
> > > Here, the term $V_M =  \sum_{i=2}^{M} \max_{\pi \in \mathcal{K}} |l_i(\pi) - l_{i-1}(\pi)|$ quantifies the total variation of the loss functions and must be characterized for each specific problem.
> > > We theoretically prove that our framework ensures $V_M$ is sublinear in the time horizon $M$ (see W2&Q2 for the full derivation). This is a crucial result, as it guarantees that the overall dynamic regret $\mathcal{R}_D$ is also sublinear. Consequently, this implies that the learner's policy converges, on average, toward the optimal expert policy.
> > >
> > > 4) Finally, we address the practical challenges of implementing FTPL-D+, which assumes access to an ideal offline optimization oracle. Since such an oracle is computationally infeasible, we approximate it using multi-step mini-batch Stochastic Gradient Descent (SGD), which serves as an effective surrogate. To further enhance training stability and mitigate the risk of converging to poor local minima, we periodically reset the network parameters. This combined strategy enables stable and effective optimization in a practical setting.

---

> > > ### Author Response · Authors · 2025-08-09
> > >
> > > We hope our response above has addressed the reviewer's concerns. If the reviewer has any further questions, please feel free to let us know.

---

### Official Review · Reviewer_MP88 · 2025-07-03

**Clarity:** 2
**Significance:** 3
**Originality:** 3
**Rating:** 5
**Confidence:** 4

**Summary:**

This paper introduces faithful dynamic imitation learning for human in the loop imitation learning. It propose a unbiased imitation learning objectives that  faithfully imitates human behavior and  contains a novel problem formulation by framing HIL as an online learning problem to tackle the non-stationary of the data distribution as agent’s policy improves.  Through experiments on the MetaDrive and CARLA simulators, the paper demonstrates that FaithDaIL achieves state-of-the-art performance in terms of safety and task success, while requiring significantly less human intervention data compared to existing HIL baselines.

**Questions:**

1. The non-stationarity you address is a common issue that also persists in general reinforcement learning (e.g., when the policy updates change the state visitation distribution). Do you believe the core ideas of your method, particularly the dynamic regret framework, could be generalized to improve sample efficiency or stability in standard RL settings beyond HIL?
2. Can you explain the rationale behind selecting the FTPL-D+ algorithm for minimizing dynamic regret? Were alternative online learning methods evaluated during this decision process? Additionally, what practical trade-offs—such as the computational overhead introduced by the ensemble—does this choice entail compared to more straightforward approaches?

**Ethical Concerns:**

["NO or VERY MINOR ethics concerns only"]

**Final Justification:**

The author's response has resolved my issue. I will keep my score and recommend accept.

**Limitations:**

yes

**Paper Formatting Concerns:**

no concern

**Quality:**

3

**Strengths And Weaknesses:**

Strength:

1. As not an expert in HIL, I think the faithful imitation objective is very interesting and the way of converting it to the tracble form using odice through convex conjugate is clever.  For me the basic idea is similar to GAIL and reminds me of IQ-learn.
2. By observing this instability, the paper employs FTPL-D+ to frame the task as an online learning problem with a dynamic regret objective. This is a much more principled approach than simply aggregating all historical data, as it allows the model not be held back by suboptimal data from early in training.
3. the claims are supported by comprehensive empirical validation. The method outperforms a suite of baselines on two challenging driving simulators. The ablation studies clearly demonstrate the importance of both core contributions, and the qualitative case studies provide convincing visual evidence of the agent's superior safety and stability.

Weakness:

1.  The FTPL-D+ framework requires maintaining an ensemble of K policies, value functions, and discriminators. This approach seems computationally expensive and may pose scalability challenges, especially as the model complexity or the length of the training horizon (M) increases.
2. The faithful imitation objective relies on a discriminator to estimate the density ratio, which serves as a proxy reward. I am not sure if this would introduce an adversarial training dynamic similar to that in GANs. The stability of such training could be a practical issue.
3. The paper would benefit from a more detailed and intuitive explanation of the FTPL-D+ algorithm. While the reference is provided, a brief, self-contained overview of why this ensemble-based, multi-interval approach is well-suited to minimizing dynamic regret would make the paper more accessible to a broader audience.

---

> ### Author Rebuttal · Authors · 2025-07-31
>
> ### W1: Computational Overhead
>  We agree that an ensemble of K policies increases computational overhead. However, based on empirical observations from our implementation, we propose two practical strategies to manage this cost.
>
>  First, in practise, we observe that the agent's policy distribution progressively converges toward the expert policy distribution, and the best-performing policies within the ensemble are usually those from recent rounds with shorter time intervals. This suggests that the full ensemble, which is required by the theoretical formulation, may not be necessary in practice. The computational cost can be significantly reduced by maintaining only a smaller subset of learners focused on more recent data, without a major impact on performance.
>
>  Second, to address scalability as the training horizon (M) grows, the process can be partitioned into sequential stages in practice. Each stage runs for a shorter horizon and "warm-starts" using the best policy from the previous one. This approach prevents the ensemble size from growing indefinitely across a very long training process. We plan to evaluate this staging strategy in future work.
>
>
>
>
>
>
> ### W2: Adversarial Training Instability
>  We thank the reviewer for this important question. We agree that training stability with a discriminator is a key concern, and we designed FaithDaIL's framework to be fundamentally different from adversarial approaches like GAIL or GANs.
>
>  Our method avoids a direct min-max game by decoupling the optimization of the policy and the discriminator. At each round, the discriminator is first trained to help compute a proxy reward and importance weights. The policy is then updated via weighted behavior cloning to imitate behaviors with higher weights, rather than trying to fool the discriminator. This sequential and non-adversarial training paradigm, inherited from the DICE family of algorithms, is inherently more stable [1].
>   Furthermore, the FTPL-D+ ensemble provides an additional layer of robustness by adaptively managing multiple learners, which naturally mitigates the impact of any single, potentially volatile, discriminator estimation. We will revise Section 4.1 to better clarify this decoupled update process.
>
> [1] DemoDICE: Offline Imitation Learning with Supplementary Imperfect Demonstrations (2022 ICLR)
>
>
>
>
> ### W3: Explanation of FTPL-D+
>    We thank the reviewer for the helpful suggestion. We will add the following explanation of the FTPL-D+ algorithm in Section 4.2.
>
>    FTPL-D+ is an ensemble variant of Follow-the-Perturbed-Leader (FTPL), designed to handle non-stationary characteristics where the loss landscape changes over time. It maintains a set of $K$ FTPL learners, each associated with a different rolling window size (e.g., covering the past $h$ rounds), to capture different temporal dynamics. This mechanism implicitly smooths out the non-stationarity of the data distribution and allows the overall strategy to adapt to varying timescales of change. Such a multi-interval ensemble is particularly well-suited for minimizing dynamic regret, as it balances short-term reactivity (via short-horizon learners) and long-term stability (via long-horizon learners). Theoretical results of FTPL-D+ [2] show that this structure achieves near-optimal dynamic regret bounds in general non-convex settings.
>
> [2] Online non-convex learning in dynamic environments (2024 NIPS)
>
>
>
>
> ### Q1：Generalization to RL
>
>    We thank the reviewer for this insightful question. We think that the core ideas of our FaithDaIL -- optimize dynamic regret to address non-stationarity distribution --- could benefit in a range of non-stationary RL environments, including：
>   - Non-stationary dynamics, due to environmental changes, multi-agent interactions, or sim-to-real domain shifts: One possible approach is to handle non-stationary environmental changes by formulating an objective to evaluate the accuracy of the dynamic transition model within the dynamic regret framework, which helps the learner adapt to shifts in the environment's underlying dynamics.
>   - Reward function drift, as seen in preference-based or feedback-driven learning: we can solve the non-stationary reward function shift by defining the expected cumulated reward as the objective of dynamic regret and solving it using a method like FTPL-D+
>   - Evolving tasks in continual reinforcement learning: we can define a sequence of task-specific loss functions, where each loss may reflect metrics like task completion performance, and incorporate them into the dynamic regret framework accordingly.
>
>
> ### Q2: Rationale of using FTPL-D+
>
>    We clarify the motivation of using FTPL-D+ as follows. In the domain of online non-convex learning, previous approaches to minimizing dynamic regret often rely on prior knowledge about the dynamics of the loss landscape or require computationally intractable sampling oracles. Therefore, they cannot be used in our HIL-IL problem due to the unknown dynamics.
>
>   In contrast, FTPL-D+ does not depend on such prior knowledge, and its underlying offline optimization oracle is much more practical—stochastic gradient descent can serve as an effective approximation in practice and often leads to near-global optima. Given these advantages, we believe FTPL-D+ is currently the most suitable and only choice for addressing our problem setting.
>
>   We clarified the computational overhead may not be a issue in practice (see Respone to W1).

---

### Official Review · Reviewer_ZERT · 2025-07-03

**Clarity:** 3
**Significance:** 4
**Originality:** 3
**Rating:** 5
**Confidence:** 2

**Summary:**

This work proposes Faithful Dynamic Imitation Learning (FaithDaIL), a framework for Human-in-the-loop imitation learning that enhances data efficiency by integrating evolving novice trajectories through dynamic regret minimization. The authors start their paper by motivating the need for their approach and discussing the drawbacks and limitations of prior work. The authors then formulate learning from active human intervention with evolving policy trajectories as a dynamic regret minimization problem. Afterward, the authors propose FaithDaIL and describe several components, including the loss function and algorithm. Finally, FaithDaIL is evaluated in two challenging driving simulators. The authors find that FaithDaIL achieves SOTA safety and task success with much-reduced human intervention.

**Questions:**

+ Please respond to the weaknesses noted above.

**Ethical Concerns:**

["NO or VERY MINOR ethics concerns only"]

**Final Justification:**

My final score for this paper is a 5: Accept. The proposed work is interesting and validated well. With the additional clarifications and findings provided during the author-reviewer discussion period, I think this paper would be a strong contribution to NeurIPS.

**Limitations:**

yes

**Quality:**

3

**Strengths And Weaknesses:**

Strengths:
+ This paper deploys their algorithm in very challenging domains. As the algorithm is able to perform well in these domains, as well as require less human intervention, this lends support to the significance of FaithDaIL.
+ The contributions in this paper are significant and theoretically grounded.
+ The experiments in this paper utilize real human data collected from participants.

Weaknesses:
- Figure 2 is difficult to interpret and could use some enhancing and further captioning. The support text is helpful.
- Could you provide further information regarding the experiment procedure and HIL experiments? Was data for HG-DAgger collected in the same manner?

---

> ### Author Rebuttal · Authors · 2025-07-31
>
> ### W1: Figure Caption
> Thanks for the comments. We will add the following descripion in the caption for interpretability.
>
> We can see that both HG-DAgger and IWR fail in senarios with both static and dynamic obstacles (deviates off-road in (b) and (e), collides in (a) and (f)), and have erratic trajectories in curving roads (c) and (d). HACO performs even worse, shows unstable behavior like wide swings in (c), (d), and (e)  and lane departures in (a) and (b). PVP showed risky maneuvers to avoid obstacles, resulting in collides in (f) and deviates in (a), (b), and (e). In contrast, FaithDaIL produces the smoothest and safest trajectories, closely aligning with expert driving across all scenarios.
>
>
>
>
>
> ### W2: Details on HIL Experiments
>  We provided the HIL experiment details in Appendix C. To improve clarity, we reorganize and revise the experiment details here. The described Human-in-the-Loop (HIL) procedure was used to collect data for HG-DAgger and all baseline methods in the same manner.
>
> Human Subject Recuitment: Three college students (aged 20–25), each holding a valid driver’s license and with at least two years of driving experience, voluntarily participated in the human-in-the-loop experiments. Participants were recruited through campus-wide advertisements and selected based on their availability, driving experience, and willingness to engage in interactive training tasks. We ensured full transparency by clearly explaining the purpose of the study, the nature of their involvement, and how the collected data would be used. Each participant signed a written informed consent form, indicating their full understanding and voluntary agreement. The study protocol was reviewed and approved by the university Institutional Review Board (IRB), ensuring compliance with ethical standards for research involving human subjects.
>
> HIL Experiments Procedure: Before commencing the experiments, all participants received detailed instructions and underwent a brief training session to familiarize themselves with the simulator interfaces and intervention mechanisms. They practiced for approximately 30 minutes until achieving proficiency, which was defined as attaining $\ge 95\%$ task success over 50 consecutive episodes. In the main human-in-the-loop (HIL) sessions,  each participant supervised a learning agent, intervening via a keyboard in MetaDrive or a Logitech G923 racing wheel in CARLA. The objective is to navigate the vehicle safely to the destination. Participants are encouraged to intervene whenever they sense potential danger or behavior misaligned with human norms. To mitigate potential order bias, the presentation sequence of algorithms (for experiments in MetaDrive), simulators, and task scenarios was randomized for each participant. Each human-in-the-loop (HIL) data collection session last approximately one hour (70 minutes for tasks in MetaDrive and 60 minutes for tasks in CARLA). Consequently, each participant dedicated a total of 5 hours to the MetaDrive experiments (which involved interacting with 5 different algorithms). For the CARLA simulator, each participant engaged with our FaithDaIL method for one 60-minute session; performance data for other baseline methods on CARLA were adopted from the findings reported in PVP.

---

> > ### Comment · Reviewer_ZERT · 2025-08-06
> >
> > Thank you for your rebuttal. After reading all reviews and author responses, I have decided to maintain my score. If the paper is accepted, please include the aforementioned details about the HIL experiments.

---

### Official Review · Reviewer_xuad · 2025-07-03

**Clarity:** 2
**Significance:** 2
**Originality:** 3
**Rating:** 4
**Confidence:** 3

**Summary:**

The paper studies the problem of imitation learning with online expert annotation, in which the learner can access the environment and generate states, meanwhile the expert can provide action for each state. The paper formulates it as a dynamic regret minimization problem, and proposes a learning approach based on the follow-the-perturbed-leader (FTPL) paradigm. Experimental results under autonomous driving benchmarks are provided to verify the effectiveness of the proposed approach.

**Questions:**

Please feel free to respond to the points in the weaknesses part.

**Ethical Concerns:**

["NO or VERY MINOR ethics concerns only"]

**Final Justification:**

My major concerns have been addressed in the rebuttal. Thus my score is raised accordingly.

**Limitations:**

The discussions of technical limitations seem to be lacking.

**Quality:**

2

**Strengths And Weaknesses:**

- Strengths:

1. The paper suggests studying dynamical regret minimization in imitation learning, which is less-studied to my knowledge. This may inspire online learning theory researchers to consider imitation learning as a practical applicational area for dynamic regret studies.

- Weaknesses:

1. The learning problem may not be properly formulated. The dynamic regret actually assumes that the optimal policy changes over time, while in the paper, the optimal policy remains the same as the oracle policy. Even though in Eq. 1, $\pi^*$ seems to be related to time index $i$, while the paper does not assume that the oracle's policy can change. Instead, the paper claims that the reason to consider dynamical regret is that the learner's policy can change. While this situation is common for all online learning problems, even when the static regret is considered. Actually, the motivation for the classical DAGGER algorithm is to consider the effect of changing learner policy.

2. In the abstract and introduction part, it is said the expert reaction delay can be an issue. While the paper seems to lack further discussions on the problem.

3. The notion of faithfulness needs further clarification. It seems that the paper utilizes this as a measurement on how good the learner can imitation the expert. But the relationship of faithfulness to the dynamic regret problem still lacks clear explanations.

4. The paper lacks theoretical analysis on regret guarantees, which is important for online learning researches.

---

> ### Author Rebuttal · Authors · 2025-07-31
>
> ### W1: Justification for Dynamic Regret
>   We clarify that our problem is not "imitation learning with online expert annotation". We study human-in-the-loop imitation learning with active human intervention, in which, human experts supervise the agent's execution, and intervene to make corrections once errors occur. The key difference of our human intervention from "online expert annotation" is that we learn from a  hybrid of expert trajectories as well as the changing agent-generated trajectories (which we call novice data), while "online expert annotation" assumes stationary sample demonstrations. *This changing distribution of imitation data,  is the key reason why we adopt dynamic regret.*  In contrast, DAgger uses static regret because its data is consistently sampled from the fixed human expert policy distribution.
>
>   In many classical dynamic‑regret formulations, the data distribution and loss function or environment dynamics  (summarized as loss landscape) change over time. As a result, the optimal policy that minimizes instantaneous loss often varies from one round to the next. However, *we argue dynamic regret is designed for problems with non-stationary loss landscape. Whether the optimal policy changes over time is not the determining factor for using dynamic regret.* [1] studied dynamic regret when optimal policy remains unchanged but  loss or data distribution changes. In this setting, the changing loss landscape introduces complexity beyond what is captured by static regret.
>
>   Therefore, although our faithful imitation objective (Section 4.1) assumes that the theoretical optimal policy is fixed, we can still formulate our problem as a dynamic regret problem because of the complexity induced by the changing distribution of imitation data.
>
> [1] On Online Optimization: Dynamic Regret Analysis of Strongly Convex and Smooth Problems (AAAI 2021)
>
>
>
>
>
>
>
>
> ### W2: Link to expert reaction delay
> We thank the reviewer for this comment. We will clarify the connection in the revision.
>
> Expert reaction delay creates suboptimal "deviation trajectories" where the agent acts incorrectly just before a human intervenes. Learning directly from this data leads to unfaithful imitation. To address this, our faithful imitation objective optimizes the KL divergence between the agent's policy and the expert's policy. Our method assigns lower proxy rewards to suboptimal trajectories, which in turn gives them lower weights during the weighted behavior cloning update. This process mitigates the negative impact of intervention delays, ensuring the agent learns to imitate expert policy faithfully.
>
>
>
>
>
>
>
>
> ### W3: faithfulness clarification
> The notion "faithfulness" is mainly about imitating human experts only, while still leveraging trajectory data from both human experts and the learning policy for data efficiency. This is to alleviate the negative impact of the suboptimial of learning policy trajectory data, and achieved by a faithful imitation objective that optimizes the difference between the behavior policy and the expert policy (see Section 4.1) with a discrimilator.
>
> We clarify that our target is to imitate a hybrid dataset consisting of human expert trajectories and the learning policy trajectories for data efficiency. This hybrid dataset brings two key callenges: 1) changing distribution and 2) deviation trajectories caused by expert reaction delay. The changing distribution is addressed by formulating the whole imitation learning problem as dynamic regret, and the effect of deviation trajectories addressed by our proposed faithful imitation objective.  This faithful imitation objective is used as a loss in dynamic regret.
>
> Furthermore, we argue that dynamic regret promotes more faithful imitation compared to static regret. Our approach allows the model to track the most recent and highest-quality policy trajectories, which progressively converge toward the expert's policy. This avoids the bias from outdated, suboptimal data that would dilute a static regret objective, thereby leading to a more faithful imitation.
>
>
>
>
>
>
> ### W4: Theoretical Analysis
>
> Thanks for the comments, we will provide the following theoretical analysis on regret guarantees. Our theoretical analysis shows that our algorithm is no-regret and can eventually converge to the optimal policy.
>
> (We encountered a rendering issue with long equations in the OpenReview rebuttal system. To address this, we have broken them down into smaller terms, which required introducing new notations (e.g., $k = i-1$) and may harm the paper's readability.)
>
> We aim to prove that our FaithDaIL is no-regret and can eventually converge to the optimal policy, i.e.,
>
> $
> \frac{1}{M} \mathbb{E}[\mathcal{R}_D] \to 0
> $
>
>
> We first give some assumption:
>
> + **Bounded Parameter Space**: The policy parameter space is effectively bounded due to regularization and optimizer constraints in practice.
> + **Lipschitz Loss**: The imitation loss $\ell_i(\pi)$ is Lipschitz in $\theta_\pi$, assuming (1) $\pi(a|s)$ is Lipschitz, (2) importance weights $\omega_i^\star(s,a)$ are bounded, and (3) $\pi(a|s)$ is bounded away from zero.
>
> Following the analysis from FTPL-D+, the **expected dynamic regret** of FaithDaIL satisfies:
>
> $
> \mathbb{E}[\mathcal{R}_D] \le \mathcal{O}\left( M^{\frac{2}{3}} (V_M + 1)^{\frac{1}{3}} \right),
> $
>
> where:
>
> - $M$ is the number of rounds,
> - $V_M =  \sum_{i=2}^{M} \max_{\pi \in \mathcal{K}} |l(\pi, \mathcal{D}_i^B, \mathcal{D}_i^H) - l(\pi,\mathcal{D}_k^B, \mathcal{D}_k^H)|$, $k=i-1$ is the total variation of the loss sequence.
>
> To establish that the dynamic regret is sublinear, **it suffices to show that $V_M$ grows sublinearly with $M$**. We formalize the implicit intervention rate $\beta_i$ and state the core assumption underlying its use:
>
> + **Definition (Implicit Intervention Rate $\beta_i$)** :
>
> $
> \beta_i = \mathbb{E}_{\pi^N_i } \left[ \frac{|\mathcal{D}_i^{H,\text{new}}|}{|\mathcal{D}_i^{H,\text{new}}| + |\mathcal{D}_i^N|} \right]
> $
>
> + **Assumption (Intervention Rate Decay)** : This reflects the intuition that as the agent improves, fewer human interventions are needed over time.
>
> $
> \lim_{i \to \infty} \beta_i = 0
> $
>
> To relate $\beta_i$ to the trajectory distribution shift, we invoke the following result (adapted from DAgger, Lemma 4.1):
>
> + **Lemma (Distribution Divergence Bound)**: With $T$ as the MDP horizon,
>
> $
> \|d_i^B - d_i^N\|_1 \le 2T\beta_i
> $
>
> We now bound the single-step loss variation $\Delta_i = \max_{\pi} | \ell_i(\pi) - \ell_{i-1}(\pi) |$.
>
> $
> |\ell_i(\pi) - \ell_{i-1}(\pi)|
> \leq |A - B| + |B - C|
> $
>
> where $k=i-1$ :
>
> $
> A = \mathbb{E}_{(s,a) \sim d_i^B}[w_i^\star(s,a) \log \pi(a|s)]
> $
>
> $
> B = \mathbb{E}_{(s,a) \sim d_k^B}[w_i^\star(s,a) \log \pi(a|s)]
> $
>
> $
> C = \mathbb{E}_{(s,a) \sim d_k^B}[w_k^\star(s,a) \log \pi(a|s)]
> $
>
>
>
> 1. **Term 1 (due to distribution shift):**：
>
> $
>    \left| A-B \right| = \left| \int_{(s,a)\in\mathcal{S}\times\mathcal{A}}[d_{i}^B(s,a) - d_{k}^B(s,a)] w_i^\star(s,a) \log \pi(a|s) ds  da  \right|
> $
>
> $
> \leq N_1 \cdot N_2 \cdot \|d_i^B - d_k^B\|_1
> $
>
> where:
>
> $
> N1 = \|w_i^\star \|_{\infty}
> $
>
> $
> N2 = \|\log \pi\|_{\infty}
> $
>
> 2. **Term 2 (due to weight shift):**：
>
> $
>    \left| B-C \right| = \left| \int_{(s,a)\in\mathcal{S}\times\mathcal{A}}[w_{i}^\star(s,a) - w_{k}^\star(s,a)] d_k^B(s,a) \log \pi(a|s) ds  da  \right|
> $
>
> $
> \leq N_2 \cdot N_3 \cdot \|w_{i}^\star - w_{k}^\star\|_1
> $
>
> where:
>
> $
> N2 = \|\log \pi\|_{\infty}
> $
>
> $
> N3 = \|d_k^B \|_{\infty}
> $
>
> **Step 1: Bounding distribution shift:** Using triangle inequality and Lemma (Distribution Divergence Bound):
>
> $
> \|d_i^B - d_k^B\|_1 \le 2T(\beta_i + \beta_k) + \|d_i^N - d_k^N\|_1
> $
>
> Assuming the novice policy changes smoothly:
>
> $\sum_{i=2}^M \|d_i^N - d_{i-1}^N\|_1 = o(M)$
>
>  and $\beta_i \to 0$, we have:
>
> $
> \sum_{i=2}^M \|d_i^B - d_{i-1}^B\|_1 = o(M)
> $
>
> **Step 2: Bounding weight shift:** Assuming that the variation in importance weights is primarily induced by the shift in behavior distributions, and that $V_i^\star$ evolves smoothly under $d_i^B$, we have:
>
> $
> \|w_i^\star - w_{i-1}^\star\|_1
> $
>
> $
> \le L_w \cdot \|d_i^B - d_{i-1}^B\|_1 + o(1)
> $
>
> Thus:
>
> $
> \sum_{i=2}^M \|w_i^\star - w_{i-1}^\star\|_1 = o(M)
> $
>
> Then under the earlier Lipschitz assumptions, we assume relevant terms of N1, N2, N3 are uniformly bounded.
>
> **Conclusion: Sublinear functional variation.**
>
> $
> V_M = \sum_{i=2}^M | \ell_i(\pi) - \ell_{i-1}(\pi) | = o(M)
> $
>
> **Final Result: No-regret Guarantee**
>
> From FTPL-D+ analysis:
>
> $
> \mathbb{E}[\mathcal{R}_D] = \mathcal{O}(M^{2/3}(V_M + 1)^{1/3}) = o(M), \quad
> \frac{1}{M} \mathbb{E}[\mathcal{R}_D] \to 0
> $
>
> Hence, **FaithDaIL is a no-regret algorithm** in non-convex dynamic environments, with the learned policy converging to the round-wise optimal.

---

> > ### Comment · Reviewer_xuad · 2025-08-05
> > **further question**
> >
> > Thanks for the responses, which address most my previous concerns. I have a further question regarding "W1: Justification for Dynamic Regret". I agree that the reason to use dynamic regret is that the target policy changes over time, which is the mixture of the expert and learner policy. However, does this mean that even though the dynamic regret can be minimized, the learner may still not be able to learn the optimal policy? Intuitively, this depends on how frequent the intervention can be conducted. While is there any way to quantitatively analyze this?

---

> ### Author Response · Authors · 2025-08-05
>
> We thank the reviewer for the positive feedback on our responses and for raising this important follow-up question. Regarding this question:
>
> We clarify that our target policy is the fixed expert policy. Specifically, we proposed a faithful imitation objective that directly sets the expert policy as the imitation target, while still leveraging mixed novice and expert data. Nevertheless, this objective can still be optimized within the dynamic regret framework, as clarified in our earlier response.
>
> As we optimize for the expert policy, minimizing the dynamic regret ensures that the learner converges to the optimal policy—*a process that does not inherently depend on the frequency of human intervention*. Our empirical results suggest that convergence can be achieved under relatively sparse interventions, which highlights the practical robustness of our approach.
>
> The reviewer's concern is exactly the limitation of prior HIL methods (IWR, HACO, and PVP), in which, the imitation objective is implicitly defined on the mixture of the expert and learner policies. And we believe quantifying human intervention frequency will be a good way to measure how good their solutions are theoretically.

---

> ### Comment · Reviewer_xuad · 2025-08-06
>
> Thanks for the reply. While I don't quite understand why the proposed approach can approach the optimal expert policy without any conditions. The optimization objective (Eq. 1 in the paper), i.e. the dynamic regret, compares the performance of the learned policy to the optimal policy up to the current round. Assume that human intervention is significantly sparse, then these temporary optimal policies can have a large gap to the ground-truth optimal policy. I would like to know if any further theoretical justifications can be provided.

---

> ### Author Response · Authors · 2025-08-06
>
> We sincerely thank the reviewer for the thoughtful follow-up. We clarify that our method can approach the optimal expert policy *conditioned on the learning-from-human-intervention setting*. In this framework, a human expert ensures safety and alignment with human intent. Specifically, whenever the agent makes an error or deviates from human preference, the expert intervenes to guide it onto a correct trajectory.
>
> This establishes a direct relationship between the learner's performance and the rate of intervention. A higher frequency of errors, which signifies a larger gap between the learner's policy and the optimal policy, necessitates more frequent interventions. Conversely,  *a significantly sparse intervention rate indicates that the learner's policy is successfully converging toward the expert's.*
>
> Here, we provide the theoretical justifications of the relationship between the round specific gap to the ground-truth optimal policy $\ell_i(\pi_i^N) - \ell_i(\pi^H)$ and intervention rate $\beta_i$ as follows:
>
> (We encountered a rendering issue with long equations in the OpenReview rebuttal system. To address this, we have broken them down into smaller terms, which required introducing new notations and may harm the paper's readability.)
>
> First, according to Eq. (8) in the paper, we have:
>
> $
> \ell_i(\pi_i^N) - \ell_i(\pi^H) = \mathbb{E}_{(s,a) \sim d_i^B} \left[ \omega_i^*(s,a) \log \frac{\pi^H(a|s)}{\pi_i^N(a|s)} \right]
> $
>
> According to the lemma in the preceding proof (see our response in W4), we have:
>
> $
> \| d_i^B - d_i^N \|_1 \leq 2T \beta_i
> $
>
> Let $f(s, a) = \omega_i^*(s, a) \log \frac{\pi^H(a|s)}{\pi_i^N(a|s)}$. It is known that the difference in expectations under two distributions can be bounded by their total variation distance:
>
> $
> |A- B| \leq \sup |f| \cdot \| d_i^B - d_i^N \|_1
> $
>
> where,
>
> $A=\mathbb{E}_{d_i^B}[f]$
>
> $B=\mathbb{E}_{d_i^N}[f]$
>
> Combining the above inequalities, we obtain:
>
> $
> \ell_i(\pi_i^N) - \ell_i(\pi^H) =  \mathbb{E}_{d_i^B}[f]
> $
>
> $
> \leq \mathbb{E}_{d_i^N}[f] + C \cdot \beta_i
> $
>
> where $C = 2T \cdot \sup |f|$ is a constant.
>
> The first term $\mathbb{E}_{d_i^N}[f]$ in the above equation measures the discrepancy between the novice policy and the expert policy during the novice's own exploration.
>
> A larger $\mathbb{E}_{d_i^N}[f]$ indicates a greater deviation from human expectations, leading to a higher intervention rate $\beta_i$ since human experts intervene whenever the novice policy makes mistakes.
>
> Next, we derive the relationship between $\mathbb{E}_{d_i^N}[f]$ and intervention rate $\beta_i$.
>
> Since intervention is determined by human subjective judgment, we can assume that for a particular human expert, there exists a psychological threshold for intervention, beyond which the expert chooses to intervene. Formally, we define *intervention event* as follows:
>
> When the divergence between the novice policy and the expert policy at a certain state $s$ exceeds a threshold $\delta$, a human intervention occurs:
>
> $
> E_{\text{int}}(s) \equiv \\{ D_{KL}(\pi^H(\cdot \mid s) \,\|\, \pi_i^N(\cdot \mid s)) > \delta \\}
> $
>
> We then relate the intervention rate $\beta_i$ to the event $E_{\text{int}}$:
>
> $
> \beta_i = \mathbb{E}_{s \sim d_i^N(s)} \[ I1 \]
> $
>
> where
> $
> I1 = \mathbb{I}\left( E_{\text{int}}(s) \right)
> $
>
> The policy gap expectation can be decomposed as:
>
> $
> \mathbb{E}_{d_i^N}[f] =  A2 + B2
> $
>
> $
> A2 = \mathbb{E}_{s \sim d_i^N(s)} \[ I1 \cdot f_s \]
> $
>
> $
> B2 = \mathbb{E}_{s \sim d_i^N(s)} \[ I2 \cdot f_s \]
> $
>
> $
> I1 = \mathbb{I}\left( E_{\text{int}}(s) \right)
> $
>
>
> $
> I2 = \mathbb{I}(\neg E_{\text{int}}(s))
> $
>
> where
>
> $
> f_s = \mathbb{E}_{a \sim \pi_i^N(\cdot \mid s)} \left[ \omega_i^*(s, a) \log \frac{\pi^H(a \mid s)}{\pi_i^N(a \mid s)} \right]
> $
>
> is the policy gap at state $s$.
>
> We provide upper bounds for both components:
>
> For states likely to trigger intervention, we assume an upper bound of $f_s$ under these states as $C_{\text{high}}$:
>
> $
> A2 \leq \beta_i \cdot C_{\text{high}}
> $
>
> For states unlikely to trigger intervention, we assume a small constant upper bound of $f_s$ under these states as $\epsilon_{\text{safe}}$:
>
> $
> B2 \leq  \epsilon_{\text{safe}} \cdot (1 - \beta_i)
> $
>
> Therefore, the term $\mathbb{E}_{d_i^N}[f]$ can be bounded as
>
> $
> \mathbb{E}_{d_i^N}[f]
> $
>
> $
> \le \beta_i \cdot C_{high} + (1-\beta_i) \cdot \epsilon_{safe}
> $
>
> $
> = (C_{high}-\epsilon_{safe}) \cdot \beta_i+\epsilon_{safe}
> $
>
> When the novice policy deviates significantly from the human expert policy, we can have $C_{high} > \epsilon_{safe}$. As the novice policy approaches the expert policy, $\epsilon_{safe}$ approaches 0.
>
> In summary, we have:
>
> $
> \ell_i(\pi_i^N) - \ell_i(\pi^H) \leq (C+C_{high}-\epsilon_{safe}) \cdot \beta_i+\epsilon_{safe}
> $
>
> That is, the loss gap between the temporary optimal policy and the the ground-truth optimal policy at each round is bounded by the intervention rate $\beta_i$. As $\beta_i \to 0$, the learner will converge to the human expert policy.

---

> > ### Comment · Reviewer_xuad · 2025-08-07
> >
> > Thanks for the detailed analysis. I think the additional theoretical results, along with the author responses, make the theoretical justification of the proposed approach clearer and more complete. I will increase my score accordingly, and I encourage to put these additional contents in the revision.

---

> > > ### Author Response · Authors · 2025-08-07
> > >
> > > We sincerely thank the reviewer for the thoughtful and encouraging feedback. We are glad to hear that the additional theoretical analysis has clarified our approach, and we will reflect these improvements in the revised submission.

---

### Note · Authors · 2025-08-13

We thank the reviewers for their insightful feedback. We are thrilled they found our work to address a **"meaningful problem"** (``DSG8``) and offer **"new insights"** (``DSG8``), with contributions deemed **"significant and theoretically grounded"** (``ZERT``) and our core method praised as **"clever"** (``MP88``). Both ``ZERT`` and ``MP88`` recommended acceptance from the start.

Our rebuttal successfully addressed the reviewers' initial concerns, providing **a new no-regret analysis** (``xuad, DSG8``) and clarifying our **methodology and key contributions** (``MP88, DSG8, ZERT``), which the reviewers kindly acknowledged.

We are also grateful for the active discussion phase. We were able to address the initial concerns of Reviewer ``xuad``, and we are encouraged that the reviewer has indicated an intention to raise the score in light of our response. The follow-up discussion with the reviewer ``DSG8`` was also productive. After acknowledging that most concerns were addressed, ``DSG8`` asked for a final clarification on the **technical novelty** of our solution to the dynamic regret problem. In our final response, we clarified that our novelty is multi-faceted:
- **Conceptual Novelty**: The primary contribution is not a new algorithm for minimizing dynamic regret, but the **formal framing of HIL-IL as an online non-convex learning problem** with non-stationary data.
- **Technical Innovation**: We creatively repurposing the Hedge-based sampling mechanism within FTPL-D+ as a novel, adaptive policy-switching mechanism for data collection.
- **Theoretical Contribution**: We provide a **new dynamic regret bound analysis** tailored specifically for our HIL-IL framework.

We are confident this response fully clarifies the novelty of our work. The entire review process has been invaluable in strengthening the paper, and we are confident in its contribution to the NeurIPS community.

---

### Decision · Program_Chairs · 2025-09-17

**Decision:**

Accept (poster)

**Comment:**

This paper studies imitation learning with online expert annotation, and formulates it as a dynamic regret minimization problem.  Experimental results under driving benchmarks were convincing to the reviewers. At the rebuttal stage the reviewers were convinced about the theoretical analysis, which strengthened the paper -- please include them in the final version.